# Differential Expression Proteins Contribute to Race-Specific Resistant Ability in Rice (*Oryza sativa* L.)

**DOI:** 10.3390/plants8020029

**Published:** 2019-01-23

**Authors:** Shiwei Ma, Shoukai Lin, Menglin Wang, Yang Zou, Huan Tao, Wei Liu, Lina Zhang, Kangjing Liang, Yufang Ai, Huaqin He

**Affiliations:** 1College of Life Sciences, Fujian Agriculture and Forestry University, Fuzhou 350002, China; mshiwei@163.com (S.M.); 1170539009@fafu.edu.cn (M.W.); 1170539007@fafu.edu.cn (Y.Z.); taoh1620@gmail.com (H.T.); weilau@fafu.edu.cn (W.L.); zhanglina209@fafu.edu.cn (L.Z.); liangkj_2005@126.com (K.L.); 2Key laboratory of Loquat Germplasm Innovation and Utilization, Putian University, Fujian Province University, Putian 351100, China; linshoukai@ptu.edu.cn

**Keywords:** rice (*Oryza sativa* L.), blast fungus (*M. grisea*), race-specific resistance, different pathogenic isolates

## Abstract

Rice blast, caused by the fungus, *Magnaporthe grisea* (*M. grisea*), lead to the decrease of rice yields widely and destructively, threatening global food security. Although many resistant genes had been isolated and identified in various rice varieties, it is still not enough to clearly understand the mechanism of race-specific resistant ability in rice, especially on the protein level. In this research, proteomic methods were employed to analyze the differentially expressed proteins (DEPs) in susceptible rice variety CO39 and its two near isogenic lines (NILs), CN-4a and CN-4b, in response to the infection of two isolates with different pathogenicity, GUY11 and 81278ZB15. A total of 50 DEPs with more than 1.5-fold reproducible change were identified. At 24 and 48 hpi of GUY11, 32 and 16 proteins in CN-4b were up-regulated, among which 16 and five were paralleled with the expression of their corresponding RNAs. Moreover, 13 of 50 DEPs were reported to be induced by *M. grisea* in previous publications. Considering the phenotypes of the three tested rice varieties, we found that 21 and 23 up-regulated proteins were responsible for the rice resistant ability to the two different blast isolates, 81278ZB15 and GUY11, respectively. Two distinct branches corresponding to GUY11 and 81278ZB15 were observed in the expression and function of the module cluster of DEPs, illuminating that the DEPs could be responsible for race-specific resistant ability in rice. In other words, DEPs in rice are involved in different patterns and functional modules’ response to different pathogenic race infection, inducing race-specific resistant ability in rice.

## 1. Introduction

Rice is the dominant staple crop in the world, and the demand for rice production is still rising with the increasing population. However, rice blast caused by the fungus (*M. grisea*) leads to the decrease of rice yields widely and destructively, threatening global food security [1,2]. Rice leaves, nodes, collars, panicles, and roots are easily infected by the pathogen at all growth stages [3]. Pesticides have been over-applied to control rice blast, resulting in globe pollution. Therefore, the use of rice cultivars with resistant ability to blast fungus has been regarded as the most economical and environmental friendly approach to control this disastrous disease [4,5,6]. However, many rice cultivars only conferred race-specific resistance, and their resistance was short-lived, presumably due to strong selection pressure for high pathogenic variability of the fungus [7,8].

Isolating and subsequently identifying resistant (R) genes from rice would help to elucidate multiple molecular mechanisms for host resistance and pathogen virulence. Over 100 R genes were mapped on the rice genome, and 35 of them have been cloned [9]. However, the rice resistant ability with the single R gene is not durable due to pathogen evolution. Breeders aim to develop new rice varieties, which could balance durable resistance with yield. Deng et al. reported that two genes, including *PigmR* and *PigmS*, on one locus worked together to confer broad-spectrum resistance to blast fungus *M. grisea* without yield penalty [10]. Li et al. identified a natural allele of a transcription factor in rice with a non-race specific resistant ability to blast that carried no observable penalty in plant growth or yield [11]. Wang et al. revealed that a transcription factor, *Ideal Plant Architecture 1* (*IPA1*), could regulate the expression of *DEP1* and *WRKY45* through phosphorylation to balance resistance with yield [12]. Although the classical gene-for-gene theory could partially account for rice race-specific resistant ability to *M. grisea* [13], only seven avirulence genes (Avr) in *M. grisea* corresponding to R genes were studied, including *Pi-ta*/*AVR-Pita* [14], *Piz-t*/*AVRPiz-t* [15], *Pik*/*AVR-Pik* [16], *Pia*/*AVR-Pia* [17], *Pi-CO39*/*AVR1-CO39* [18], *Pi54*/*AVR-Pi54* [19], and *Pii*/*AVR-Pii* [16].

Proteomics, one of the most important research methods in the post-genomics era, has been widely used to study the plant response mechanism to various stresses, especially diseases. Proteomics analysis of resistant and susceptible soybean varieties infected by *Phytophthora sojae* indicated that 30 and 20 differential expression proteins (DEPs) related to metabolism, energy regulation and protein storage, and degradation were discovered in a resistant and susceptible line, respectively [20]. After *Xanthomonas oryzae* pv *oryzicola* infection, 32 proteins concerned with signal transduction, pathogenesis, and cell metabolism were up-regulated in rice leaves, and 23 of them were supported by the microarray data [21]. Ventelon-Debout et al. found 40 and 24 DEPs existed in resistant and susceptible rice varieties in response to the infection of rice yellow mettle virus (RYMV), respectively [22].

However, the relationship between the expression level of blast-induced proteins and the resistant ability of rice plants in response to the infection of different blast fungal isolates was not fully illuminated. In this study, proteomic methods were employed to analyze the DEPs in three rice varieties, which share the same genetic background, but show different resistant abilities, after inoculation with two different virulence isolates for 24 and 48 hours. The analysis on the clustering and functional module of DEPs revealed the different response mechanism of rice plants to different fungal isolates.

## 2. Results

### 2.1. Phenotype of CO39 and its two NILs in Response to Infection of Different Fungal Isolates

At the seventh day of blast fungus infection, lesion types were observed on the leaves of the three test rice varieties (Figure 1a). There was no lesion on the leaves of CN-4b under GUY11 infection, indicating that it was resistant to blast isolate GUY11. However, the other two varieties, CO39 and CN-4a, were susceptible to GUY11. In response to 81278ZB15 infection, typical spindle lesions only appeared on the leaves of CO39, but not on CN-4a and CN-4b, meaning that CO39 was susceptible, but CN-4a and CN-4b were resistant to blast isolate 81278ZB15 (Figure 1a). Furthermore, the analysis of the lesion area showed similar results (Figure 1b).

### 2.2. DEPs in Rice in Response to the Infection of Different Fungal Isolates

In this study, the 2-DE method was employed to screen the DEPs in rice leaves in response to the blast infection at 24 and 48 h post-inoculation (hpi). Protein profiles with three replicates were acquired and visualized with Coomassie staining (Figure 2 and Appendix A). More than 1000 protein spots were detected on each gel within p*I* 3–10 and MW 14.4–116.0 KDa, and most spots were distributed within p*I* 4–7. The differential expressing protein spots with more than 1.5-fold reproducible change in the relative abundance between the inoculation treatment and the control were detected using PDQuest and collected in one stained gel (Figure 2c). Based on their spot intensities, 50 protein spots were selected for identification by using MALDI-TOF-TOF/MS and results are shown in Table 1.

The above phenotype results showed that both CN-4a and CN-4b were resistant to 81278ZB15, thus the up-regulated proteins in these two varieties in response to 81278ZB15 infection were considered to contribute to rice resistance to fungal isolate 81278ZB15. A total of 21 proteins (spots 2, 3, 5, 7–9, 16, 20, 25–27, 29, 30, 32, 37, 39, 40, 43, 44, 47) were up-regulated in CN-4a or CN-4b in response to 81278ZB15 infection, among which four (spots 16, 32, 43, 47) and seven (spots 8, 14, 16, 25, 27, 37, 44) proteins were up-regulated simultaneously in these two varieties at 24 hpi and 48 hpi (Figure 3a). On the other hand, we also found that three proteins (spots 16, 43, 44) in CN-4a and two proteins (spots 16, 32) in CN-4b were induced to increase the expression level both at 24 hpi and 48 hpi.

Among the three test rice varieties, only CN-4b was resistant to blast fungus GUY11, thus the up-regulated proteins in CN-4b might be responsible for the resistance to GUY11. The expression abundance of 23 proteins (spots 2, 4, 7, 14, 16, 18–20, 23–27, 31–32, 38–39, 41, 43–45, 48–49) were induced to increase in CN-4b in response to GUY11 infection, among which 18 (spots 2, 4, 7, 14, 18, 20, 23, 25–27, 31–32, 38–39, 41, 44–45, 49) at 24 hpi and 12 (spots 4, 7, 16, 19–20, 24–26, 41, 43, 45, 48) at 48 hpi (Figure 3b). The expression levels of seven proteins (spots 4, 7, 20, 25–26, 41, 45) in CN-4b were enhanced both at 24 hpi and 48 hpi. 

Furthermore, three proteins (spots 2, 20, 32) in CN-4b were up-regulated at 24 hpi both of the two test fungal isolates, while the other three proteins (spots 16, 25, 26) were up-regulated at 48 hpi of both of the two fungal isolates (Figure 3c). At 24 hpi, both of the two fungal isolates, and three of the same proteins (spots 7, 44, and 32) in CN-4a and CN-4b were induced to increase the expression abundance, while at 48 hpi, three of the same proteins (spots 43, 16, and 25).

### 2.3. RNA Expression of DEPs in Rice in Response to Different Fungal Isolate Infection

The DEPs in rice in response to different fungal isolate infection were partially supported by RNA-seq data (unpublished) (Figure 4). At 24 hpi of GUY11, the RNA expression abundance of 16 among 32 DEPs in CN-4b increased (Figure 4a). These proteins were elongation factor G (spot 2), glycine dehydrogenase (spot 17), glyceraldehyde-3-phosphate dehydrogenase B (spot 20), fructose-bisphosphate aldolase (spot 23), class III peroxidase (spot 24), copper/zinc superoxide dismutase (spot 26), photosystem II stability/assembly factor HCF136 (spot 27), acid phosphatase (spot 29), photosystem II oxygen-evolving complex protein 1 (spot 33), nucleoside diphosphate kinase (spot 34), haloacid dehalogenase-like hydrolases (spot 35), chitinase (spot 39), NAD-dependent epimerase/dehydratase family protein (spot 40), beta-glucanase (spot 41), photosystem II oxygen-evolving enhancer protein 2 (spot 44), and serine-glyoxylate aminotransaminase (spot 48). Similarly, 16 DEPs in CN-4b were up-regulated at 48 hpi of GUY11, among which five spots’ RNA expression abundance increased correspondingly (Figure 4b). These proteins contained heat shock 70 kDa protein (spot 4), Zn-dependent oligopeptidases (spot 7), S-adenosylmethionine synthase 2 (spot 19), acid phosphatase (spot 29), and chitinase (spot 39).

### 2.4. Clustering Analysis on the Expression Pattern of DEPs in Rice in Response to Different Fungal Isolate Infection

The clustering analysis on the expression pattern of 50 DEPs was implemented by TBtools, which is shown in Figure 5. The clustering separated all treatments into two distinct clusters corresponding to the two different pathogenic isolates. The GUY11 cluster was clearly divided into two groups, one of which was the resistant branch, CN-4b, and the other was the susceptible branch, including CO39 and CN-4a. However, in the 81278ZB15 cluster, the susceptible variety, CO39, was not separated independently from the resistant varieties, but the resistant variety, CN-4b, at 24 hpi and 48 hpi of 81278ZB15 was clustered together. According to the expression abundance, all DEPs were also divided into two clusters.

### 2.5. Functional Module of DEPs in Rice in Response to Different Fungal Isolate Infection

The co-expression network of the above 50 DEPs were analyzed by using WGCNA. Seven functional modules were identified (Figure 6a) and marked as ME 1–7, which were related to amino acid metabolism, photorespiration, photosynthesis, oxidative stress, protein biosynthesis and modification, antioxidation, and energy metabolism, respectively. The absolute of *Pearson Cor* between ME 1–7 and fungal blast were 0.84 (*p* = 0.001), 0.82 (*p* = 0.001), 0.90 (*p* = 0.0001), 0.55 (*p* = 0.01), 0.56 (*p* = 0.04), 0.66 (*p* = 0.02), and 0.73 (*p* = 0.001) orderly. Subsequently, the seven functional modules were clustered, and all treatments were also grouped into two big branches according to different pathogenic isolates (Figure 6b). At 24 hpi of GUY11, the expression level of functional module ME 4, ME 5, and ME 6 in CN-4b were up-regulated more than 2-fold compared to the other two varieties, whereas the expression abundance of functional module ME 1, ME 4, and ME 5 in CN-4b were higher than the other two varieties at 48 hpi of GUY11. At 24 hpi of 81278ZB15, ME 3, ME 5, and ME 6 with an increased expression level were found in CN-4a and CN-4b compared to CO39, while ME 1, ME 3, ME 4, and ME 7 were observed to have higher expression than CO39 at 48 hpi of 81278ZB15.

## 3. Discussion

C101PKT (CN-4a) and C105TTP-4L-23 (CN-4b) with single blast resistant genes were developed by backcrossing resistant donor rice cultivars, Pai-Kan-Tao and Tetep, to the recurrent parent, CO39, respectively. Furthermore, the genetic studies showed that the resistant gene in CN-4a is allelic to that in CN-4b [23]. The pathogen inoculation results showed that both of the two NILs were resistant to blast isolate 81278ZB15, and only CN-4b was resistant to blast isolate GUY11, whereas CO39 was susceptible to both of the two blast isolates. The previous research reported that these two fungal isolates had different virulence [24]. Thus, in this study, these three rice varieties with different blast fungus resistant abilities and the two fungal isolates with different virulence were employed as materials to reveal the protein expression pattern in rice in response to different blast isolates’ infection by comparing their DEPs.

A total of 50 proteins in CO39 and its two NILs were found to be up-regulated in response to the two blast isolates’ infection. Thirteen proteins were reported to be induced by *M. grisea* in a previous research work [25,26,27], including heat shock 70 kDa protein (spot 4), NADH dehydrogenase subunit G (spot 12), NADP-dependent malic enzyme (spot 15), class III peroxidase (spot 24), glutathione S-transferase (spot 30), guanine nucleotide-binding protein subunit β-like protein (spot 31), photosystem II oxygen-evolving complex protein 1 (spot 33), L-ascorbate peroxidase (spot 36), chitinase (spot 39), dehydratase family protein (spot 40), β-glucanase (spot 41), photosystem II oxygen-evolving enhancer protein 2 (spot 44), and dihydrolipoamide dehydrogenase family protein (spot 46). Sasaki et al. also found that class III peroxidase (spot 24) was up-regulated vastly after the infection of blast fungus and its high expression was related to the defense reaction, the response to blast fungus, and basal resistance [28]. Nishizawa et al. confirmed that the up-regulation and constitutive expression of chitinase (spot 39) enhanced rice resistant ability to blast fungus [29]. Nakashima et al. revealed that guanine nucleotide-binding protein subunit β-like protein (spot 31) was up-regulated and interacted with the Rac 1 immune protein complex to improve blast resistant ability for rice [30]. However, some DEPs might be the key molecule in rice to resist blast fungus, but this not yet been reported previously. For instance, ferritin (spot 43) in CN-4b was up-regulated at 24 hpi, which was the critical response time for rice plants to resist blast fungus [31]. *Arabidopsis* plants with ferritin knocked out were sensitive to excessive iron, were deficient in growth and flowering, and showed an accumulation of the reactive oxide group [32]. Therefore, we believe that the increasing expression level of ferritin in CN-4b at 24 hpi might balance the plant growth and its resistance to fungus. 

A total of 21 proteins (spots 2, 3, 5, 7–9, 16, 20, 25–27, 29, 30, 32, 37, 39, 40, 43, 44, 47) were up-regulated in response to 81278ZB15 infection and 23 proteins (spots 2, 4, 7, 14, 16, 18–20, 23–27, 31–32, 38–39, 41, 43–45, 48–49) were induced to increase in response to GUY11 infection (Appendix A). It could be inferred that those 21 and 23 proteins were related to the resistance capacity against 81278ZB15 and GUY11, respectively. Moreover, six proteins-spots 2, 20, 32, 16, 25, and 26 (Appendix A) in CN-4b were up-regulated at 24 and 48 hpi of both two blast isolates, respectively, hinting that these proteins might contribute to a broad-spectrum resistant ability in CN-4b to blast fungus.

Clustering analysis on the expression pattern of DEPs has been widely used to find a related expression module. Salekdeh et al. discovered that protein clusters in different rice varieties’ response to drought stress in the same expression pattern and then they revealed the identical hereditary in rice in response to drought [33]. In this research, DEPs’ expression levels were clustered exactly into two branches corresponding to the two pathogenic isolates, indicating that DEPs’ cluster could reflect the race-specific resistant ability in rice. Meanwhile, compared to the other two varieties, DEPs in CN-4b at 24 and 48 hpi of both two blast isolates were always clustered together with a characteristic module, inferring that CN-4b had a unique basal defense system to blast fungus. 

The functional modules of DEPs were grouped into two branches correlated to the two pathogenic isolates. This result was consistent with that in the DEP cluster analysis. The photosynthesis-related module ME 3 in resistant varieties (CN-4a and CN-4b) was expressed higher than that in the susceptible one CO39 in response to the infection of 81278ZB15, because its photosynthesis was sensitive to various stresses and it decreased after pathogen infection [34]. When the plant was infected by pathogens, free radicals and excessive peroxides were produced and this resulted in damage to cells [35,36]. The increasing expression level of oxidative stress (ME 4 and 6) in CN-4b might confer resistance to GUY11 by removing the free radicals and regulating peroxide accumulation [11].

In summary, 50 proteins were found to be up-regulated in rice in response to different fungal isolate infection, and the expression pattern and functional modules of these 50 DEPs were clustered into two branches corresponding to the two blast isolates, indicating that different DEPs contribute to the race-specific resistant ability in rice.

## 4. Materials and Methods

### 4.1. Plant Materials and Fungus Inoculation

A rice variety, CO39, susceptible to rice blast and its 2 NILs, including C101PKT (CN-4a) and C105TTP-4L-23 (CN-4b), were employed as materials in this research work. Two blast fungus isolates, *GUY11* and *81278ZB15*, with different virulence were used to infect rice plants [24]. Rice seedlings at the stage of four and half emerging leaves were spray-inoculated with 25 mL blast fungus spore suspensions (1 × 10^5^ spores/mL suspended in 0.01% Tween-20) in a growth chamber for 24 h in darkness at 26 °C, and were subsequently kept at 12 h/12 h (day/night), 26 °C and 90% relative humidity. At 24 and 48 hpi, rice leaves infected with blast fungus were harvested in liquid nitrogen and then stored at −80 °C for further analysis. Lesion types on leaves were observed after 7 days of infection [37] and ImageJ was used to analyze the percentages of lesion areas. Three replicates of each treatment were performed and the mean value was used for further analysis.

### 4.2. Two-Dimensional Electrophoresis (2-DE) and Spots Selection

Proteins were extracted from frozen leaves according to He and Li [38]. The total protein concentrations in samples were determined following Bradford [39], using bovine serum albumin as the standard. 2-DE on the total proteins was run using the method introduced by Chen et al. with minor modifications [40]. The first dimensional IEF was performed by Protean IEF Cell (Bio-Rad, USA) by using IPG strips (17 cm, pH3-10, nonlinear, Bio-Rad, USA). Subsequently, the focused IPG strips were immediately equilibrated in equilibration buffer I (375 mM Tris-HCl, pH 8.8, 6 M urea, 20% glycerol, 2% SDS, 2% DTT) and equilibration buffer II (375 mM Tris-HCl, pH 8.8, 6 M urea, 20% glycerol, 2% SDS, 2.5% iodoacetamide) for 15 min orderly. Then, the 2-D electrophoresis was run on 12% sodium dodecyl sulfate-polyacrylamide electrophoresis (SDS-PAGE) gels. Following electrophoresis, the 2-D SDS-PAGE gels were stained with Coomassie blue dye. The stained gels were scanned with an Image Scanner III system (GE Healthcare), and then images were analyzed using PDQuest software (version 8.01, Bio-Rad, Hercules, CA, USA). Triplicate gels were used for each treatment and the mean value represented the content of spots. The protein spots with more than 1.5-fold reproducible change in relative abundance, as well as being statistically significant (using Student’s *t*-test, at *p* < 0.05), were considered to be DEPs and subjected to MALDI-TOF-TOF/MS analysis.

### 4.3. MALDI-TOF-TOF/MS Analysis and Protein Identification

The interest protein spots were identified by MALDI-TOF-TOF/MS according to Chen’s method [40]. Protein spots from the stained gel were digested with sequencing-grade trypsin (Promega, USA), and then the resulting peptides were desalted with C18 ZipTips (Millipore, USA), and mixed with 5 mg/ml alpha-cyanocinnamic acid in 70% acetonitrile and 0.1% trifluoroacetic acid. ABI 4700 proteomics analyzer (Applied Biosystems, USA) were employed to detect mass spectra in both the MS and MS/MS modes. Data were analyzed for protein identification with MASCOT 2.2 software (http://www.matrixscience.com/) based on the non-redundant protein set from NCBInr and Swissport. Typical search parameters were set as: Mass tolerance, 1.0 Da; missed cleavages, 1; enzyme, trypsin; fixed modifications, carbamido-methylation; variable modification, oxidation (M); taxonomy, Viridiplantae. The displayed message of protein spots contains the NCBInr number and its corresponding IRGSP number, the description, molecular weight (MW), isoelectric point (p*I*), MASCOT expect, peptides matched, and the cover.

### 4.4. RNA Extraction, Library Construction

The total RNA of each leaves’ sample was isolated using the Trizol Kit (Promega, Madison, WI, USA) following the manufacturer’s instructions and treated with RNase-free DNase I (Takara Bio, Shiga, Japan) for 30 min at 37 °C to remove residual DNA. RNA quality was verified using a 2100 Bioanalyzer (Agilent Technologies, Santa Clara, CA, USA) and was also checked by RNase free agarose gel electrophoresis. The transcriptome assembly library as a reference library was constructed by mixing equal amounts of RNA from the above samples. Then, Poly (A) mRNA was isolated using oligo-dT beads (Qiagen, Hilden, Germany). All mRNA was broken into short fragments by adding fragmentation buffer. First-strand cDNA was generated using random hexamer-primed reverse transcription, followed by the synthesis of the second-strand cDNA using RNase H and DNA polymerase I. The cDNA fragments purified using a QIAquick PCR extraction kit were washed with EB buffer for end reparation poly(A) addition and ligated to sequencing adapters. Following agarose gel electrophoresis and extraction of cDNA from gels, the cDNA fragments were purified and enriched by PCR to construct the final cDNA library.

### 4.5. Transcriptome Sequencing and de novo Assembly 

The cDNA library was sequenced on the Illumina sequencing platform (IlluminaHiSeq™ 4000) using the paired-end technology by Gene Denovo Co. (Guangzhou, China). The obtained raw reads were pre-processed by Trimmomatic (http://www.usadellab.org/cms/?page=trimmomatic) to remove the adaptor sequences, resulting in clean reads. Clean reads from each sample were de novo assembled by the TopHat software and then the bam files were programmed to obtain transcripts by Cufflinks [41]. For the three biological replicates of every treatment, transcripts from the same treatment were merged using Cuffmerge [41]. These three programs above all used the Nipponbare genome (http://rice.plantbiology.msu.edu) as a reference genome. 

### 4.6. Identification of Differentially Expressed Genes

The Cuffdiff [41] software was employed to evaluate the differential gene expression between any two samples. The gene abundance differences between sample pairs were calculated based on the ratio of the FPKM values. The significance of gene expression differences was calculated using the FDR (false discovery rate) control method to justify the p-value. Here, only genes with an absolute value of log2 ratio ≥ 0.76 (fold-change ≥ 1.7) and a FDR significance score ≤ 0.001 were used for subsequent analysis.

### 4.7. Clustering Analysis of the Expression Pattern of DEPs

To find the unknown field and deduce the function of unknown genes based on known genes, TBtools was used to analyze the expression pattern of DEPs with the following parameters: Log Scale Base = 2.0, LogWith = 1.0, Cluster Rows, Cluster cols, Show Value and Show on the Value [42].

### 4.8. Co-Expression Network and Functional Module Analysis on DEPs

Weighted Gene Co-expression Network Analysis (WGCNA) constructed the co-expression network based on the correlation coefficient of expression pattern [43]. The WGCNA 1.46 program in R 3.2.0 was employed to analyze expression data of DEPs with the following parameters: MinSize = 5, deepsplit = 3, power = 4. Cytoscape (version 3.2.1) software was used to visualize the functional modules.

## Figures and Tables

**Figure 1 plants-08-00029-f001:**
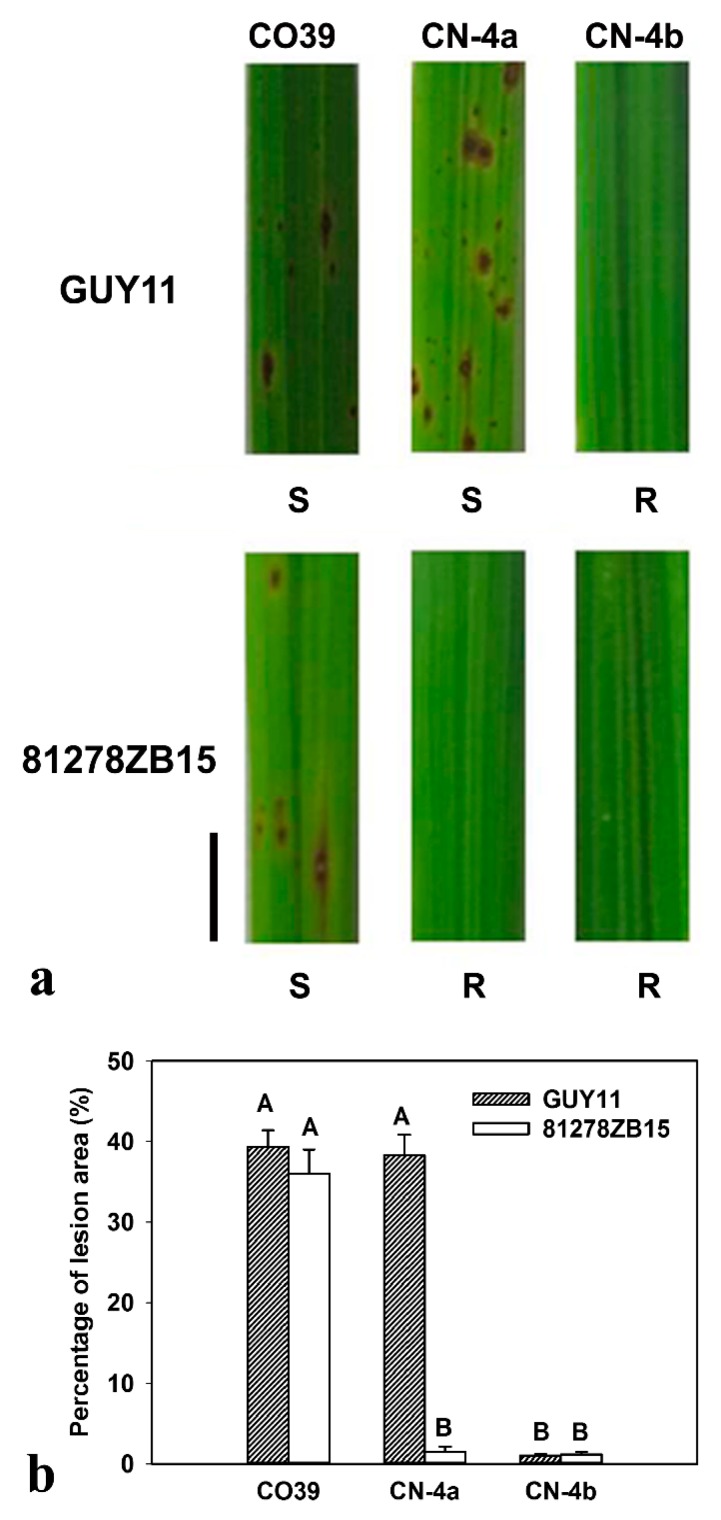
Phenotypes of CO39 and its two NILs inoculated with different fungal isolates, GUY11 and 81278ZB15. (**a**) Lesions on rice leaves at 7 days post inoculation (dpi) with blast fungus. At the seventh day, there is no lesion on the leaves of CN-4b, but full lesion on the leaves CO39 after infection by both of the two isolates. Lesions appeared on the leaves of CN-4a after infection by GUY11, but disappeared after infection by 81278ZB15. S: Susceptible, R: Resistance, Bar = 1.0 cm. (**b**) The percentage of lesion area at 7 dpi with GUY11 and 81278ZB15. Means labeled with different letters indicate significant difference at the 1% level via a Tukey-Kramer test for multiple comparisons.

**Figure 2 plants-08-00029-f002:**
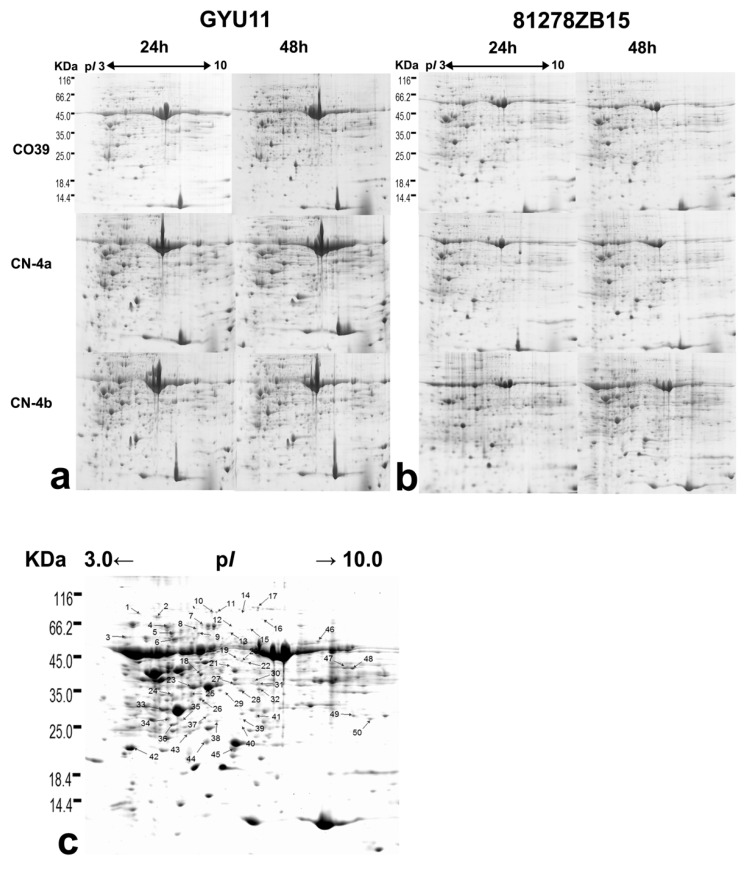
2-DE maps of CO39 and its two NILs in response to infection of GUY11 and 81278ZB15. (**a**) 2-DE maps of the three test varieties at 24 and 48 hpi of GUY11. (**b**) 2-DE maps of the three test varieties at 24 and 48 hpi of 81278ZB15. (**c**) Profile of DEPs in the three test rice cultivars in response to blast fungus infection. Number on C corresponds to different expression protein spots.

**Figure 3 plants-08-00029-f003:**
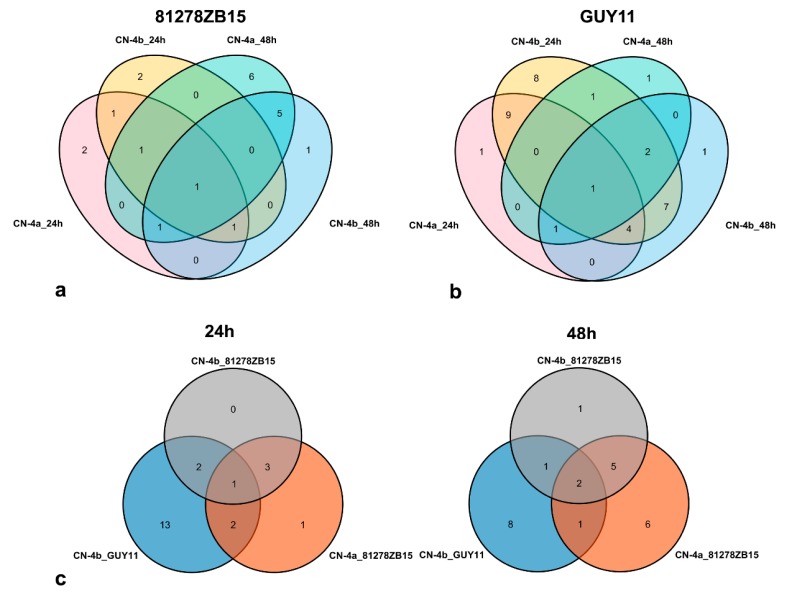
Veen of DEPs in rice at 24 and 48 hpi of GUY11 and 81278ZB15. (**a**) Veen of DEPs in CN-4a and CN-4b at 24 and 48 hpi of 81278ZB15. (**b**) Veen of DEPs in CN-4a and CN-4b at 24 and 48 hpi of GUY11. (**c**) Veen of DEPs in CN-4a and CN-4b at 24 hpi (Left) and 48 hpi (Right) of both GUY11 and 81278ZB15. CN-4a_24 h or _48 h: CN-4a at 24 or 48 hpi, CN-4b_24 h or _48 h: CN-4b at 24 or 48 hpi, CN-4a_81278ZB15 or _GUY11: CN-4a in response to the infection of 81278ZB15 or GUY11, CN-4b_81278ZB15 or _GUY11: CN-4b in response to the infection of 81278ZB15 or GUY11.

**Figure 4 plants-08-00029-f004:**
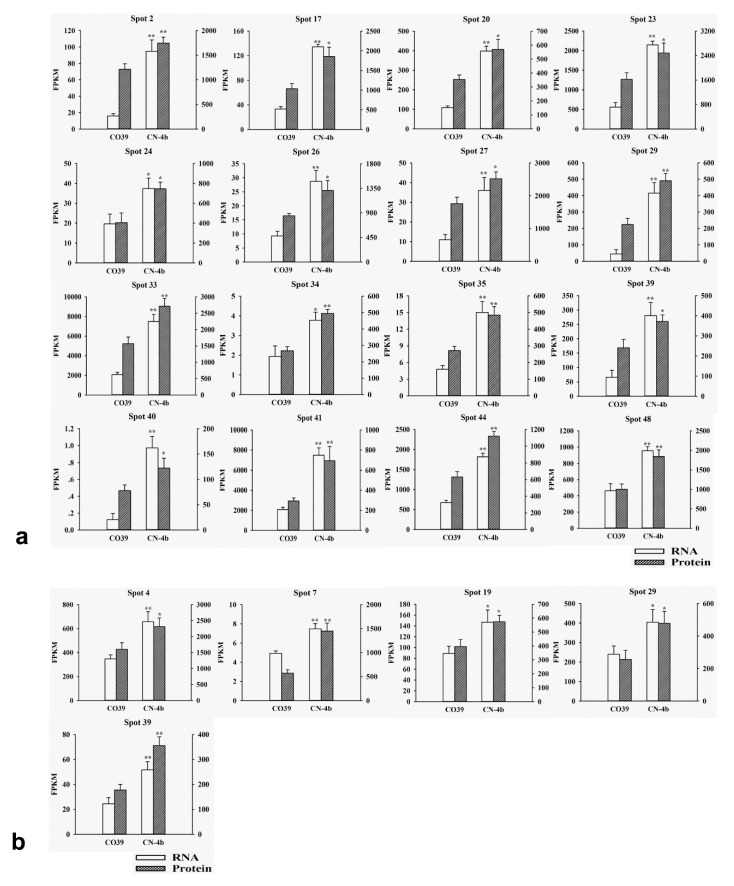
RNA abundance of DEPs in CN-4b in response to GUY11 infection. (**a**) RNA and its protein expression abundance of 16 DEPs in CN-4b at 24 hpi of GUY11. These 16 DEPs’ RNA expression abundance was up-regulated at 24 hpi of GUY11, as well as their corresponding protein abundance. (**b**) RNA and its protein expression abundance of 5 DEPs in CN-4b at 48 hpi of GUY11. The five DEPs’ RNA expression abundance was up-regulated at 48 hpi of GUY11, as well as their corresponding protein abundance. * indicates significance at the 0.05 level, while ** shows significance at the 0.01 level (using student’s *t*-test).

**Figure 5 plants-08-00029-f005:**
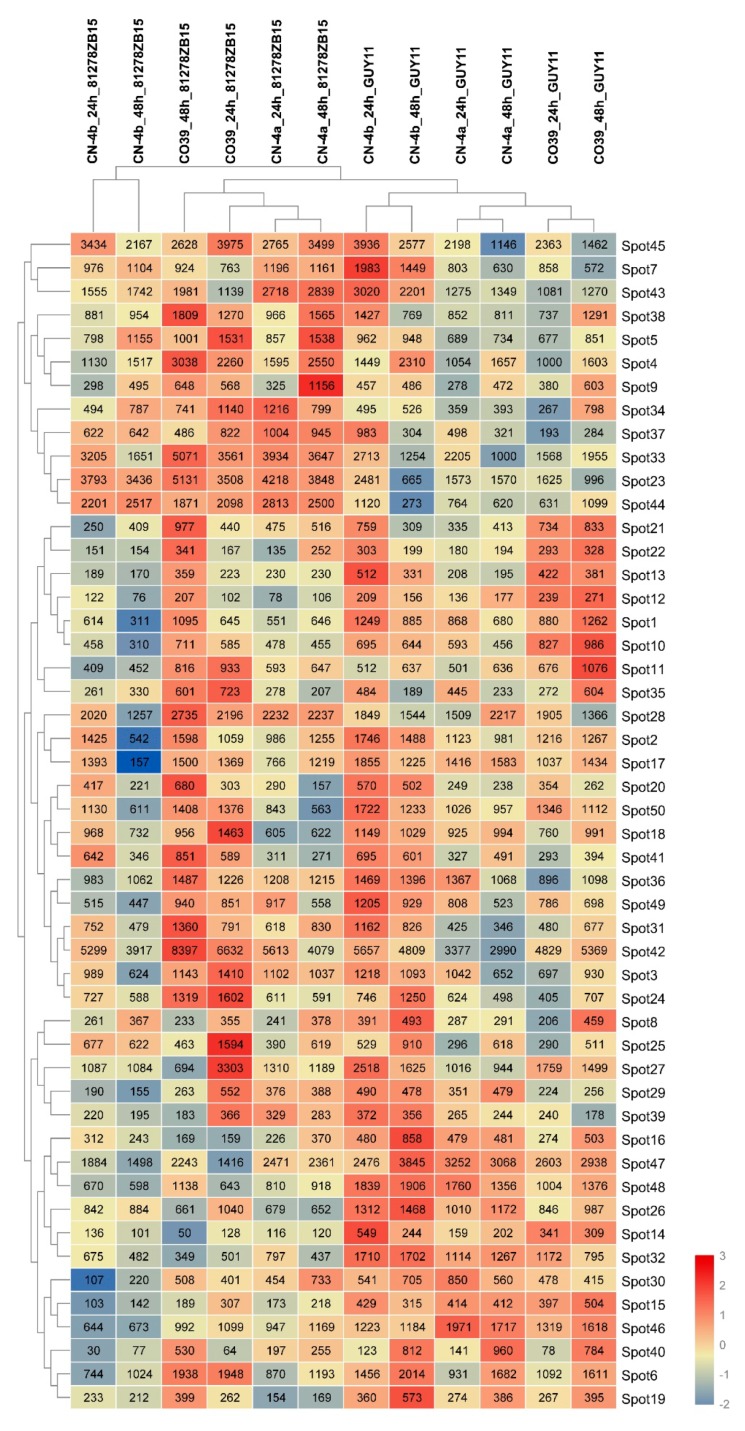
Heat map of expression pattern of DEPs in the three test rice cultivars at 24 and 48 hpi of different fungal isolates. According to different samples, the DEPs were grouped into two clusters, one was GUY11 and the other was 81278ZB15. Based on the similarity of the expression level, protein spots were separated into two clusters, the upper one contained spots from spot 45 to spot 44. CN-4a_24h_81278ZB15 or _GUY11: CN-4a at 24 hpi of 81278ZB15 or GUY11, CN-4b_24h_81278ZB15 or _GUY11: CN-4b at 24 hpi of 81278ZB15 or GUY11, CO39_24h_81278ZB15 or _GUY11: CO39 at 24 hpi of 81278ZB15 or GUY11. The same below.

**Figure 6 plants-08-00029-f006:**
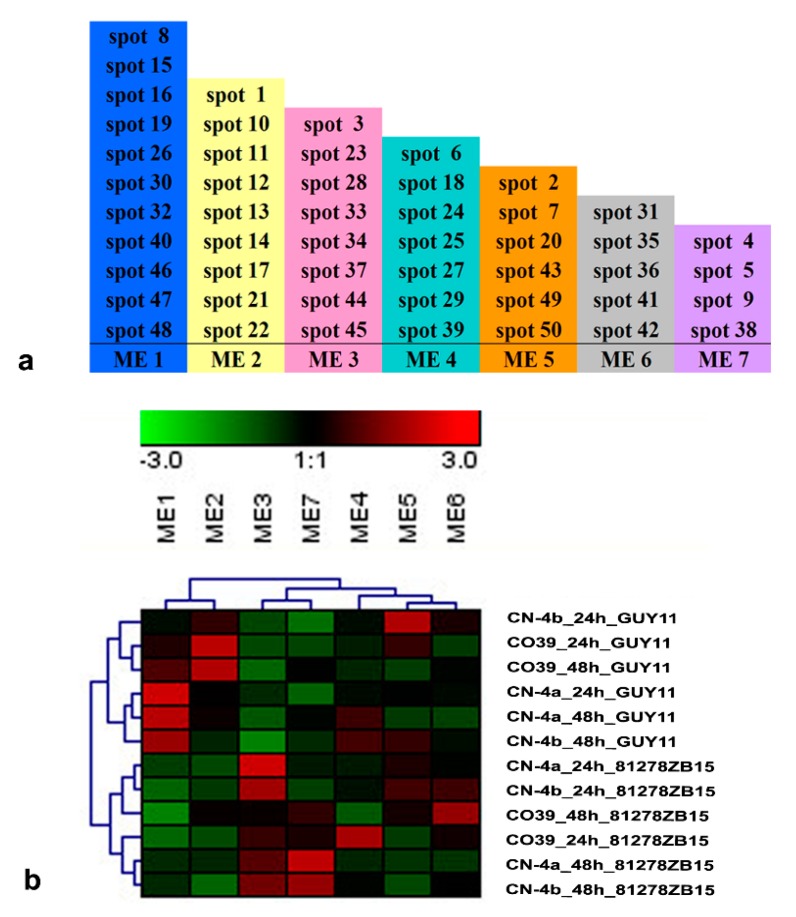
Functional modules and their clusters of DEPs in the test rice cultivars in response to blast fungus infection. (**a**) DEPs in rice in response to different fungal isolate infection belonging to seven functional modules. (**b**) The cluster of functional modules in rice in response to different fungal isolate infection. ME 1–7 were related to amino acid metabolism, photorespiration, photosynthesis, oxidative stress, protein biosynthesis and modification, antioxidation, and energy metabolism, respectively. According to different samples, the functional modules were grouped into two clusters, one was GUY11 and the other was 81278ZB15.

**Table 1 plants-08-00029-t001:** Identifications of DEPs in rice in response to different fungal isolate infection.

No ^a^	Gi ^b^	Protein Description	MW ^c^ (KDa)	p*I* ^d^	Peptides matched	Cov (%) ^e^
1	gi|115479725	Heat shock 90 kDa protein	75.0	8.94	10	18
2	gi|115459660	Elongation factor G (EF-G) family	42.1	5.47	12	41
3	gi|573913556	Histone-lysine N-methyltransferase ATX4-like	105.0	7.65	3	5
4	gi|115486793	Heat shock 70 kDa protein	71.5	5.10	7	16
5	gi|115469362	V-type proton ATPase catalytic subunit A	68.7	5.20	16	39
6	gi|115466004	Similar to 60 kDa chaperonin (Protein Cpn60)	64.3	5.60	10	25
7	gi|115450022	Zn-dependent oligopeptidases	86.5	5.76	9	14
8	gi|115448989	Molecular chaperone DnaK	73.1	5.49	13	26
9	gi|115440691	2,3-bisphosphoglycerate-independent phosphoglycerate mutase	61.0	5.42	6	14
10	gi|347602486	Chaperone protein ClpC1, chloroplastic	101.9	6.14	33	41
11	gi|115487910	Chaperone protein ClpC2, chloroplastic	102.1	6.62	9	12
12	gi|115454943	NADH dehydrogenase subunit G	82.1	5.86	15	33
13	gi|115458140	Similar to Glycyl-tRNA synthetase	32.8	5.47	5	22
14	gi|75225211	Putative aconitate hydratase	98.6	5.67	18	22
15	gi|115439655	NADP-dependent malic enzyme	65.6	8.59	9	23
16	gi|115489652	5-methyltetrahydropteroyltriglutamate--homocysteine methyltransferase	84.9	5.93	12	24
17	gi|115468926	glycine dehydrogenase	97.9	5.98	9	10
18	gi|115448531	Glutamine synthetase	39.4	5.51	5	17
19	gi|3024122	S-adenosylmethionine synthase 2	43.3	5.68	10	31
20	gi|115450493	Glyceraldehyde-3-phosphate dehydrogenase B	47.5	6.22	8	23
21	gi|115482032	GDP-mannose-3,5-epimerase (GME)-like	43.1	5.75	7	25
22	gi|115461951	S-denosylmethionine synthetase 1	43.6	5.74	14	37
23	gi|115484401	Fructose-bisphosphate aldolase	42.2	6.38	8	35
24	gi|115445243	Similar to Class III peroxidase	34.8	5.32	4	23
25	gi|115482534	Cytoplasmic malate dehydrogenase	35.9	5.75	10	36
26	gi|115477837	Copper/zinc superoxide dismutase (SOD)	21.4	5.79	5	41
27	gi|937924719	Similar to Photosystem II stability/assembly factor HCF136, chloroplastic	42.9	9.05	8	27
28	gi|125537696	ferredoxin-NADP(+) reductase	41.1	8.26	4	10
29	gi|115435022	Similar to Acid phosphatase	33.2	6.33	10	30
30	gi|115450565	Similar to Glutathione S-transferase GSTF14	35.7	7.04	6	27
31	gi|115439261	Similar to Guanine nucleotide-binding protein subunit beta-like protein	36.7	5.97	8	34
32	gi|115443911	Similar to NADPH-dependent mannose 6-phosphate reductase	42.1	8.16	6	16
33	gi|115436780	Similar to Photosystem II oxygen-evolving complex protein 1	34.8	4.96	3	21
34	gi|115488968	Nucleoside diphosphate kinase	23.6	9.51	3	11
35	gi|115453797	Haloacid dehalogenase-like hydrolases	34.1	8.36	6	25
36	gi|115474285	L-ascorbate peroxidase	27.2	5.21	7	45
37	gi|218189522	Phage shock protein A (IM30)	35.1	9.08	5	16
38	gi|115447465	Similar to ATP-dependent Caseinolytic protease (ClpP) proteolytic subunit	32.1	6.71	7	22
39	gi|297604125	Similar to Chitinase	32.8	6.08	10	26
40	gi|115461679	NAD-dependent epimerase/dehydratase family protein	31.4	9.13	6	31
41	gi|115463555	Beta-glucanase	34.7	5.92	5	23
42	gi|115446541	Similar to 2-cys peroxiredoxin BAS1	28.3	5.67	3	42
43	gi|115486898	Eukaryotic ferritins	28.3	5.47	6	29
44	gi|115470529	photosystem II oxygen-evolving enhancer protein 2	27.1	8.66	5	28
45	gi|115476760	Cupin_1; Similar to Germin-like protein 1	22.0	6.01	1	10
46	gi|115436320	Dihydrolipoamide dehydrogenase family protein	53.0	7.21	5	18
47	gi|115460656	Similar to Aminomethyltransferase	44.3	8.53	7	19
48	gi|115477148	Serine-glyoxylate aminotransaminase	44.4	8.19	4	18
49	gi|115478779	Outer mitochondrial membrane protein porin (Voltage-dependent anion-selective channel protein) (VDAC)	29.2	7.07	6	28
50	gi|115460338	Heme-dependent peroxidases	38.4	8.67	14	42

a: Numbers correspond to the 2-DE gel in Figure 2. b: Protein Gi number in NCBI. c,d: Theoretical molecular weight and p*I* value. e: Sequence coverage percentage.

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
