# Peer review of "Differential Expression Proteins Contribute to Race-Specific Resistant Ability in Rice (Oryza sativa L.)"

_plants, 2019, doi:10.3390/plants8020029_

Round 1
Reviewer 1 Report
Minor revisions (highlighted in yellow) were done on the manuscript which is hereby uploaded.

Author Response
Response to Reviewer 1’s commentsComment 1: Minor revisions (highlighted in yellow) were done on the manuscript which is hereby uploaded.
Response: We have revised all the highlighted points based on the reviewer 1’s suggestion. Thanks for reviewing the manuscript and offering suggestions.

Reviewer 2 Report
Comments on Ma et al. 2018,
In this manuscript the Authors used a proteomic approach to identify a number of proteins differentially expressed in a susceptible rice variety and two near isogenic lines upon challenging with two M. grisea isolates with different pathogenicity. This approach allowed the Authors to identify 50 proteins differentially expressed proteins, with some of them potentially implicated in resistance mechanism towards this pathogen.
The topic is of interest for the journal readership since it provides useful information about race specific resistance ability in rice. I have only one main concern related to the provided transcriptomic data, which were used to validate the proteomic results.
In this context, the Authors decided to present transcriptomic data only at the same time points used for proteomic analyses. This is surprising since up-regulation of genes in most cases precedes the increase in protein content. Therefore, at least one earlier time should be included (i.e. 12 hpi for transcripts encoding proteins upregulated at 24 hpi and 24 hpi for transcripts encoding proteins upregulated at 48 hpi.
Minor points:
1. Check typos and the overall quality of English.
Author Response
Response to Reviewer 2’s comments
Comment 1: In this context, the Authors decided to present transcriptomic data only at the same time points used for proteomic analyses. This is surprising since up-regulation of genes in most cases precedes the increase in protein content. Therefore, at least one earlier time should be included (i.e. 12 hpi for transcripts encoding proteins upregulated at 24 hpi and 24 hpi for transcripts encoding proteins upregulated at 48 hpi.
Minor points: Check typos and the overall quality of English.
Response: Although the up-regulation of genes precedes the increase in protein content, the change trend of RNA expression level is usually consistent with the protein abundance. At the same time point, the RNA expression level is able to reflect the expressing abundance of related protein to some extent [1].
We have checked typos and the overall quality of English carefully.
Reference
1.Tian D, Yang L, Chen Z et al. Proteomic analysis of the defense response to Magnaporthe oryzae in rice harboring the blast resistance gene Piz-t. Rice 2018;11:47.

Reviewer 3 Report
The manuscript by Ma et al. is focused in the identification of differentially expressed proteins in three rice genotypes associated to race-specific resistance against rice blast. Although some interesting results have been obtained, I have important concerns on the experimental design and the presentation and discussion of the results.
1. Authors state that different genotypes are resistant/susceptible to blast isolates based in phenotypical observations after 7 days of infection. To support this conclusion, more time points and any quantification of leaf damage or fungal amount should have been provided. This point is very important since authors use 24h and 48h post inoculation as time points to compare proteomes.
2. The proteomic results are confusing. Control gels from plants not treated should have been provided to an accurate comparison. Besides, it is unclear the criteria to select protein spots, if they come from each specific comparison treatment/control or from a global comparison treatment/control. In any case, a quantification of their relative abundance in the three genotypes is necessary to fully understand the proteins that are up-regulated specifically in any genotype after fungal inoculation.
3. Functional modules do not added significant information. Most of them are composed by proteins related to different physiological processes and discussion about them is weak. Authors focus in ME4 and 6 as specific for CN-4b resistance to GUY11 when actually ME5 seems the most up-regulated module in CN-4b.
4. Several sentences in the discussion are based in very weak assumptions:
p.16 l.1: “These indicated that the genetic background of the two NILs is the same as CO39 except the resistant genes”
p.16 l.10: “Total 50 proteins in CO39 and its 2 NILs were found to be up-regulated under the 2 blast isolates infection, over half of which were confirmed by RNA-sequencing result” (RNA-seq expression data do not confirm protein expression)
p.17 l. 19 “Meanwhile, compared to the other 2 varieties, DEPs in CN-4b at 24 and 48 hpi of both 2 blast isolates were always clustered together with a characteristic module, inferring that CN-4b had a different basal defense system to blast fungus from the other 2 rice varieties”
Author Response
Response to Reviewer 3’s comments
Comment 1: Authors state that different genotypes are resistant/susceptible to blast isolates based in phenotypical observations after 7 days of infection. To support this conclusion, more time points and any quantification of leaf damage or fungal amount should have been provided. This point is very important since authors use 24h and 48h post inoculation as time points to compare proteomes.
Response: The different resistant/susceptible phenotype of the 3 NILs used in this research work to the different blast isolates had been reported in many publications [1] and our previous research [2]. All the researches indicated that 24 hpi was the critical time points for rice defense to blast fungus [2, 3]. Therefore, we want to compare the race-specific up-regulating proteins in rice in response to 24h and 48h of blast isolate infection. We harvested the rice plant samples at 24h and 48h for proteomic research work and the rest of seedlings were used to confirm resistant/susceptible phenotype at 7 days of infection. According to the reviewer’s comment, we had provided more detail in the Material and Methods and Results part, such as the fungal amount and the percentage of lesion area.
Comment 2: The proteomic results are confusing. Control gels from plants not treated should have been provided to an accurate comparison. Besides, it is unclear the criteria to select protein spots, if they come from each specific comparison treatment/control or from a global comparison treatment/control. In any case, a quantification of their relative abundance in the three genotypes is necessary to fully understand the proteins that are up-regulated specifically in any genotype after fungal inoculation.
Response: We have submitted the control gels in supplementary files. The global comparison treatment/control was done in our study. Triplicate gels were used for each treatment and the mean value represented the content of spots.
Comment 3: Functional modules do not added significant information. Most of them are composed by proteins related to different physiological processes and discussion about them is weak. Authors focus in ME4 and 6 as specific for CN-4b resistance to GUY11 when actually ME5 seems the most up-regulated module in CN-4b.
Response: WGCNA was used to analyze the functional modules with strict criteria. We focused on the race-specific functional module, such as ME 4 and ME 6. The expression level of functional module ME 4, ME 5 and ME 6 in CN-4b were up-regulated at 24 hpi of GUY11, and ME 3, ME 5 and ME 6 with increased expression level were found in both CN-4a and CN-4b. Although ME 5 was the most up-regulated module in CN-4b, it was also increased in response to 81278ZB15 in CN-4a. Thereby ME 5 might be resistant to both GUY11 and 81278ZB15.
Comment 4: Several sentences in the discussion are based in very weak assumptions:
p.16 l.1: “These indicated that the genetic background of the two NILs is the same as CO39 except the resistant genes”
Response: Because the two NILs were obtained by saturated backcrossing using CO39 as recurrent parent line, we could believe the two NILs had the same genetic background CO39 except the resistant genes.
p.16 l.10: “Total 50 proteins in CO39 and its 2 NILs were found to be up-regulated under the 2 blast isolates infection, over half of which were confirmed by RNA-sequencing result” (RNA-seq expression data do not confirm protein expression)
Response: We have revised the sentence “Total 50 proteins in CO39 and its 2 NILs were found to be up-regulated under the 2 blast isolates infection, over half (18/32 and 9/16) of which were confirmed by RNA-sequencing result.”
p.17 l. 19 “Meanwhile, compared to the other 2 varieties, DEPs in CN-4b at 24 and 48 hpi of both 2 blast isolates were always clustered together with a characteristic module, inferring that CN-4b had a different basal defense system to blast fungus from the other 2 rice varieties”
Response: We have revised the sentence.
References
1.Hua WB, Guo LU, Ming LW et al. Genetic Analysis and Molecular Marker of Avr-Pi1,Avr-Pi2 and Avr-Pi4a of Magnaporthe grisea. Yi Chuan Xue Bao 2002;29:820-826.
2.Shoukai L. Meta analysis on the blast-resistant QTLs and the excavation and expression of the blast-resistant genes in rice genome [D]. Fujian Agriculture And Forestry University, 2016.
3.Bagnaresi P, Biselli C, Orrù L et al. Comparative transcriptome profiling of the early response to Magnaporthe oryzae in durable resistant vs susceptible rice (Oryza sativa L.) genotypes. Plos One 2012;7:e51609.

Reviewer 4 Report
The manuscript deals with an interesting case of study: race-specific genes associated with the resistance of rice to different strains of Magnaporthe grisea. The research was conducted upon a proteomics approach, with the results combined with a transcriptomics analysis. A half of the differentially expressed proteins (DEPs) were confirmed using RNA-Seq; a correlation plot could better show this outcome. Overall, the manuscript is written very well, and correctly organized. Also, the Discussion section seems appropriate to me.
Therefore, I recommend accepting the manuscript for publication with only minor revisions.
Some minor comments are the following:
P4 L11: pv. (non italics)
P16 L4-5: The sentence should be changed as follows: 'The previous research reported that these 2 fungal isolates had different virulence [24].'
P16 L6-7: fungal isolates with different virulence
P21 L4: the construction of RNA libraries should be described, or it should be reported the external service for it.
Author Response
Response to Reviewer 4’s comments
Comment 1: Some minor comments are the following:
P4 L11: pv. (non italics)
P16 L4-5: The sentence should be changed as follows: 'The previous research reported that these 2 fungal isolates had different virulence.
P16 L6-7: fungal isolates with different virulence
P21 L4: the construction of RNA libraries should be described, or it should be reported the external service for it.
Response: We have revised the comments and described the steps of RNA libraries in materials and methods. Thanks for your revision and suggestions.

Reviewer 5 Report
The authors conduct proteome analysis using both susceptivle and resisatnce rice cultivirs aginast rice blast fungi. I think this work is interesitng and the quality is also alsmost sufficient for publication.
But I feel that it is a bit too much to say that differential expression of proteins between susceptible and resistance rice contribute to rice defense (as described in the title and abstract). To say this, it is necessary to show that the genes are required for defense by using the corresponding rice mutants. I dont think it is necessary to use the rice mutants in this work.
Instead, I strongly suggest to change the sentences to avoid the overstatment.
Author Response
Response to Reviewer 5’s comments
Comment: The authors conduct proteome analysis using both susceptible and resistance rice cultivars against rice blast fungi. I think this work is interesting and the quality is also almost sufficient for publication.
But I feel that it is a bit too much to say that differential expression of proteins between susceptible and resistance rice contribute to rice defense (as described in the title and abstract). To say this, it is necessary to show that the genes are required for defense by using the corresponding rice mutants. I don’t think it is necessary to use the rice mutants in this work.
Instead, I strongly suggest changing the sentences to avoid the overstatement.
Response: Based on the proteomic research results, we found the differential expression proteins between susceptible and resistance rice genotypes under the infection of different blast isolates, therefore we reached a conclusion that the differential expressing proteins contributed to rice defense. Of course, we revised the sentences to avoid overstatement according to the reviewer’s comment.

Round 2
Reviewer 2 Report
In my previous report, I found a weak correlation between proteins and transcripts abundance. This could be due to the fact that the Authors used the same time points for both the analyses. On this basis, I suggested the Authors to include an earlier time point (12 h for genes up-regulated at 24 h and 24 h for gene up-regulated at 48 h).
The Authors didn’t receive this suggestion but provided only a previous study in which a similar approach was used.
Furthermore, at page 10 and 11 of the manuscript they claim that the RNA expression abundance paralleled with protein levels either at 24 and 48 h. This is not valid for all the spots, since spots 4 and 7 at 24 h and spots 20, 25, 43, 48 at 48 h did not show significant differences with the CO39 susceptible line. I suggest the Authors to change these sentences and the legend of Fig. 4 and provide a convincing explanation of the discrepancy between gene transcript/protein levels (2 out 18 at 24 h and 4 out 9 at 48 h).
Author Response
Response to Review 2’s comments
Comments:
In my previous report, I found a weak correlation between proteins and transcripts abundance. This could be due to the fact that the Authors used the same time points for both the analyses. On this basis, I suggested the Authors to include an earlier time point (12 h for genes up-regulated at 24 h and 24 h for gene up-regulated at 48 h).
The Authors didn’t receive this suggestion but provided only a previous study in which a similar approach was used.
Furthermore, at page 10 and 11 of the manuscript they claim that the RNA expression abundance paralleled with protein levels either at 24 and 48 h. This is not valid for all the spots, since spots 4 and 7 at 24 h and spots 20, 25, 43, 48 at 48 h did not show significant differences with the CO39 susceptible line. I suggest the Authors to change these sentences and the legend of Fig. 4 and provide a convincing explanation of the discrepancy between gene transcript/protein levels (2 out 18 at 24 h and 4 out 9 at 48 h).
Response: We have considered the reviewers’ previous suggestion. It might be better to analyze RNA at the earlier time point (12h). However it is not so rigorous to only supplement the RNA-seq in response to the 12h infection. It should be better to use the same rice seedling materials to analyze the RNA-seq and proteome. We thought that it was no necessary to rerun our study and reform all data. As the reviewers’ suggestion, the asynchronous of RNA and protein expression could have influence on their correlation. However, the final protein level could be affected by multiple factors, such as the modification and degradation. Furthermore, there was similar approach used before.
Although the RNA expression level of spots 4, 7, 20, 25, 43, 48 increased slightly without significance, its protein level improved significantly. We mainly focused on the similar increased trend of the RNA and protein level, so we claim that the RNA expression abundance paralleled with protein levels. To avoid the inaccurate description, we have abandoned the interpretation of these 6 proteins and revised the Fig. 4.

Reviewer 3 Report
Authors have addressed some of my concerns conveniently, as the relationship between DEP and phenotypical issues. However, I find serious weaknesses on the interpretation of the results obtained. As previously commented, functional modules do not add significant information, since these modules contain a mix of proteins with diverse functionality. In addition, the utility of spot numbers is limited. It is very difficult to follow which spot is associated with which fungus/genotype interaction. Thus, discussion section is weak, and one of the main goals of this study, to associate DEP to rice resistance, has not been fully achieved. A detailed table including spot number, protein description and protein quantification in rice lines after fungal treatment at 24 and 48h would help to really understand how DEP are related to plant resistance/susceptibility and to make a more informative discussion.
In addition, some sentences have not been correctly changed:
- You cannot claim that the genetic background of the two NILs is the same as CO39 except the resistant genes.
- RNA-seq expression data do not confirm protein expression. They are independent quantifications that are not always linked.
Author Response
Response to Review 3’s comments
Authors have addressed some of my concerns conveniently, as the relationship between DEP and phenotypical issues. However, I find serious weaknesses on the interpretation of the results obtained.
Comment 1: As previously commented, functional modules do not add significant information, since these modules contain a mix of proteins with diverse functionality.
Response: Although we have concerned the significant information suggested by the reviewer previously, there was no parameter in WGCNA results to descript the significance of the modules. WGCNA analysis was based on the correlation of genes’ expression level, thus proteins with similar expression and different function might belong to the same module. If some proteins show similar expression pattern, these proteins will get together in the same module. For the small number proteins in every module, it was inaccurate to implement GO or KEGG analysis to identify whether proteins within one module enriched together or not. After the 7 modules obtained, we could analyze the correlation between modules and fungal isolates and supply this result in the manuscript.
Comment 2: In addition, the utility of spot numbers is limited. It is very difficult to follow which spot is associated with which fungus/genotype interaction. Thus, discussion section is weak, and one of the main goals of this study, to associate DEP to rice resistance, has not been fully achieved. A detailed table including spot number, protein description and protein quantification in rice lines after fungal treatment at 24 and 48h would help to really understand how DEP are related to plant resistance/susceptibility and to make a more informative discussion.
Response: We have added the detailed table in supplementary file Table S1 and make a discussion.
In addition, some sentences have not been correctly changed:
Comment 3: You cannot claim that the genetic background of the two NILs is the same as CO39 except the resistant genes.
Response: We have abandoned this interpretation.
Comment 4: RNA-seq expression data do not confirm protein expression. They are independent quantifications that are not always linked.
Response: We have revised the sentences to ‘Protein level was paralleled with the RNA-seq data.’

Round 3
Reviewer 2 Report
On the basis of Authors' response to my previous comments, the manuscript is now publishable on Plants.
Author Response
Response to Reviewer 2’s comments
Comment: On the basis of Authors' response to my previous comments, the manuscript is now publishable on Plants.
Response: Thanks for your patient revisions.

Reviewer 3 Report
Authors have addressed most of my concerns conveniently, and the information now included in the manuscript has improved the comprehensiveness of the results obtained. Only two minor points:
p. 18, lines 261-262 It could be inferred that those 21 proteins and 23 proteins were responsible related to for the resistance capacity against to 81278ZB15 and GUY11
p. 19, lines 277, 281
photosynthesis-related module ME 3
The increasing expression level of modules mainly related to oxidative stress and antioxidant modules (ME 4 and 6)
Author Response
Response to Reviewer 3’s comments
Authors have addressed most of my concerns conveniently, and the information now included in the manuscript has improved the comprehensiveness of the results obtained. Only two minor points:
Comment: p. 18, lines 261-262 It could be inferred that those 21 proteins and 23 proteins were responsible related to for the resistance capacity against to 81278ZB15 and GUY11
p. 19, lines 277, 281
photosynthesis-related module ME 3
The increasing expression level of modules mainly related to oxidative stress and antioxidant modules (ME 4 and 6)
Response: We have revised these points one by one suggested by the reviewer. Thanks for your significant revisions.
